# Regulation of *Jacobaea vulgaris* by varied cutting and restoration measures

**Henrike Wiggering** [ID]*, **Tim Diekötter, Tobias W. Donath**

Department of Landscape Ecology, Institute for Natural Resource Conservation, Kiel University, Kiel, Germany

* hmoehler@ecology.uni-kiel.de

**Data Availability Statement:** All relevant data are within the paper and its Supporting Information files.

**Funding:** This work was supported by the Stiftung Naturschutz Schleswig-Holstein (http://www.

## Abstract

The growth of the noxious grassland weed *Jacobaea vulgaris* Gaertn. in pastures is a threat to grazing animals. This is especially true when it dominates vegetation cover, which often occurs on non-intensively used pastures that are managed for nature-conservation, to maintain and promote biodiversity. Thus, we wanted to find management techniques to reduce *J. vulgaris* without harming the floral biodiversity on the pastures. We tested six different mechanical and cultural methods to reduce the presence and spread of *J. vulgaris*. Seven study sites in Northern Germany (Schleswig-Holstein) were treated with tilling and seeding (1), tilling and hay transfer (2), mowing twice within bloom (3), mowing before seed set and combinations of mowing and seeding with a slit drill (5) or by hand (6). Our results show that cutting within the bloom of the plant at the end of June and again four weeks later, when the plant is in its second bloom was the only treatment leading to a significant reduction in population growth rate without reducing surrounding plant species richness. The study reveals that management of *J. vulgaris* in non-intensively used pastures is possible, while preserving species-rich grasslands.

## Introduction

*Jacobaea vulgaris* is a widespread noxious grassland weed native to Eurasia, and invasive in North America, New Zealand, and Australia [1]. As the plant's pyrrolizidine alkaloids pose a health risk to cattle when consumed [2], the control of *J. vulgaris* is a primary management goal of many farmers and in some countries even prescribed in their legislation [3]. In intensive grasslands, high fertilizer input, high cutting frequency, and chemical weed management precludes the occurrence of *J. vulgaris*. Its control is challenging, however, in low input grasslands that are managed for high plant and animal diversity, yet are also prone to severe *J. vulgaris* spread [4].

Several studies have aimed to find ways to manage *J. vulgaris* [3, 5–8]. Some of these focused on chemical control measures [3], intensified fertilization [4, 9] or the introduction of antagonists to reduce *J. vulgaris* density [10]. While these measures can be very effective, their negative side-effects precludes their widespread application, especially in non-intensively managed species-rich grasslands [4, 11] or are more promising in the weed's non-native range [6].

stiftungsland.de). The funders had no role in study design, data collection and analysis, decision to publish, or preparation of the manuscript.

**Competing interests:** The authors have declared that no competing interests exist.

Studies on alternative *J. vulgaris* management measures, that are compatible with nature conservation goals, are scarce [12].

To date, research on the effectiveness of cutting on *J. vulgaris* populations in grasslands is limited [12], and so far ambiguous in its findings. While Siegrist-Maag [13] showed that frequent cutting decreases *J. vulgaris* abundance and fertility, other studies find that the species can re-grow within a few weeks after being damaged or may switch to vegetative reproduction forming multiple rosettes [2, 14]. Thus, slashing or mowing per se may not be sufficient for weed control [15].

In order to increase the effectiveness of cutting in suppressing *J. vulgaris*, the frequency and timing of cutting is essential because the proportion of nutrients and energy invested in different plant parts varies among life-stages [16, 17]. As *J. vulgaris* grows back quickly after cutting [14], a second cut may be necessary before the second bloom. While this cutting regime may prevent generative reproduction, it may induce vegetative reproduction, clonal growth or a switch to a perennial life cycle [1]. Since *J. vulgaris* usually dies off after seed production [18], cutting before seed dispersal may be another option that may not only prevent vegetative re-growth but also seed dispersal.

In addition to an adapted cutting regime, sowing other species can suppress weeds [19]. According to the biotic resistance hypotheses [20], increased plant diversity may also increase plant community resistance against future invasion [21]. Several studies have shown that the invasion of problematic plant species can be prevented through increasing plant species diversity [e.g. 22, 23].

If species enrichment proves to be an effective measure in weed control, it could be combined with the re-establishment of species-rich grassland communities, which have declined dramatically [24]. Various techniques of species enrichment are commonly applied in restoration [25]. While seeding seed mixtures is frequently applied for the re-creation of grasslands [25], transferring freshly cut, seed-containing biomass from species-rich grasslands (green hay) is another effective method to restore grasslands [26]. When green hay is transferred, disturbance of an existing sward has been shown to enhance seedling establishment [27]. Under high seed pressure from *J. vulgaris*, however, sward disturbance will also enhance the weed's establishment [9]; thus, slot drilling and broadcast sowing without sward disturbance may be promising techniques [28] to combine with the adapted cutting regimes to reduce establishment and growth of *J. vulgaris*. Therefore, we also applied a combination of cutting regimes and species-enrichment measures, where the cutting regime is thought to weaken *J. vulgaris* individuals and the increased plant diversity reduces new establishment.

Contrary to deliberate actions against *J. vulgaris*, there are also indications that waiting for its natural disappearance might be another promising management approach [14, 29, 30]. Harper [1], observed a hump-shaped population development of *J. vulgaris*, with a population boom followed by sudden decline. In addition, Bezemer et al. [29] reported a natural decrease in *J. vulgaris*. Nonetheless, management to reduce *J. vulgaris* is essential to avoid threat to domestic animals.

The objective of our study was to find the optimal treatment, timing, and frequency for grassland management on non-intensive pastures that leads to a maximal weed reduction and a minimal loss of co-occurring vegetation. We studied six different management options for five years.

We addressed the following research questions:

1. Which treatment is most effective in reducing the abundance and population growth rate of *J. vulgaris*?

2. What are the effects of the different treatments on the co-occurring grassland vegetation?

## Material and methods

### Study species

*Jacobaea vulgaris* (Asteraceae; syn. *Senecio jacobaea* L.) is native to Eurasia, but invasive in many other countries [1]. Despite being regarded as a character species of mesophilic pastures [31] in its native range, its poisonousness leads to its rating as a noxious pasture weed worldwide [3]. It commonly occurs in open disturbed sites, such as ruderal sites, fallow land, and temperate non-intensively managed grasslands [1, 2].

*J. vulgaris* is a mostly monocarpic perennial herb (Fig 2). After seed germination in late summer or autumn, the rosette overwinters and flowers the next year, with a flowering peak at the end of July. Flowering can be delayed if site conditions are not optimal or if first flowers are damaged by herbivory or cutting [14].

*J. vulgaris* has several biological traits supporting its spread and hampering its regulation. It produces thousands of wind-dispersed seeds and grows quickly at suitable sites, making it a typical pioneer species of ruderal or disturbed sites [1]. Seeds show a high germination rate (about 80%) and are long-term persistent [32]. The plant tolerates disturbance during different life stages, e.g. an early cut of the inflorescence does not preclude post-ripening of vital seeds [33] and disturbance or injury to the vegetative bud promotes regrowth through roots and crown buds [3].

### Study sites

In 2015, we began management experiments on seven pastures where *J. vulgaris* was present. Sites were situated in two geographical regions (Moraine, Hill land) in Northern Germany (see Fig 1 and S1 Fig, S1 Table) on sandy soils [34].

While five out of the seven sites were formerly used as arable land, the two remaining were managed as grassland. The current nature-conservation grazing scheme applied by the managing nature conservation organization aims at maintaining and promoting biodiversity [35]. At all sites, *J. vulgaris* abundance was very high with an average of 60 ± 5 *J. vulgaris* plants per square meter (excluding seedlings) at the experiment's start.

### Study design

Six different grassland management schemes, two measures of tilling and species enrichment, two cutting treatments, and two combinations of seeding (as a measure of species-enrichment) and cutting, against a control with no measures applied (Table 1), were assessed in a randomized block design. On each of the seven study sites, each of the six treatments was applied within an area of 90 m$^2$ (block). All study sites were under non-intensive grazing during the study. To minimize edge effects, *J. vulgaris* and the co-occurring vegetation were assessed within three permanently marked nine m$^2$ plots in the central part of each block.

In both *biodiversity* treatments, the plots were tilled with a rotary tiller to a depth of 30 cm prior to seed addition in autumn 2015. For the *biodiversity 1* treatment, 5 g/m$^2$ of the "fertile meadow mixture" (Rieger-Hoffmann seed supplier) containing 20 species (30% forbs 70% grasses) commonly found and collected in northeast Germany (see S2 Table) were sown and rolled. For the *biodiversity 2* treatment, freshly cut plant material was transferred from species-rich donor-sites nearby. Thereafter, cattle were fenced out of the seed-enriched area at least three weeks to allow seedling establishment. For the *combination* treatments, 1.5 g/m$^2$ of the seed mixture mentioned above (see S2 Table) was sown by slit drill (*combination 1*) or by hand in two subsequent years (2015, 2016; *combination 2*) after mowing the standing vegetation. Sowing took place in September and October 2015 and in October 2016 for *combination 2*. At

**Table 1. Treatments.**

| Name | Disturbance | Seeding |
|------|-------------|---------|
| Biodiversity 1 | Tilling in 2015, then left untouched in following years | broadcast-seeding in 2015 |
| Biodiversity 2 | Tilling 2015, then left untouched in following years | green hay [26] spread over the ground in 2015 |
| Flower cut | Flowers cut before each bloom (end of June and beginning of August) in every year (2015–2019) | no |
| Seed cut | Flowers cut before seeds disperse (mid/end of July) in every year (2015–2019) | no |
| Combination 1 | Mowing 2015, then left untouched in following years | drill-seeding in 2015 |
| Combination 2 | Mowing for whole study period (2015–2019 mid/end of July) | broadcast-seeding in first two years (2015–2016) |
| Control | No treatment, still grazed as all other treatments | no |

the second sowing date the *combination 2* received some additional seeds of other characteristic grassland species. Cuttings remained on the plots. Mowing took place between June and August every year from 2015 to 2019. The *flower cut* treatment was cut in late June and before the second blooming peaked. The *seed cut* treatment was cut before seed dispersal in mid to late July. Cutting of the *combination 2* treatment in late July was continued annually until the end of the experiment in July 2019.

## Data sampling

Within the central one square meter of the permanently marked plots, the position of each *J. vulgaris* individual were recorded every year between 2015 and 2019. Recorded *J. vulgaris*

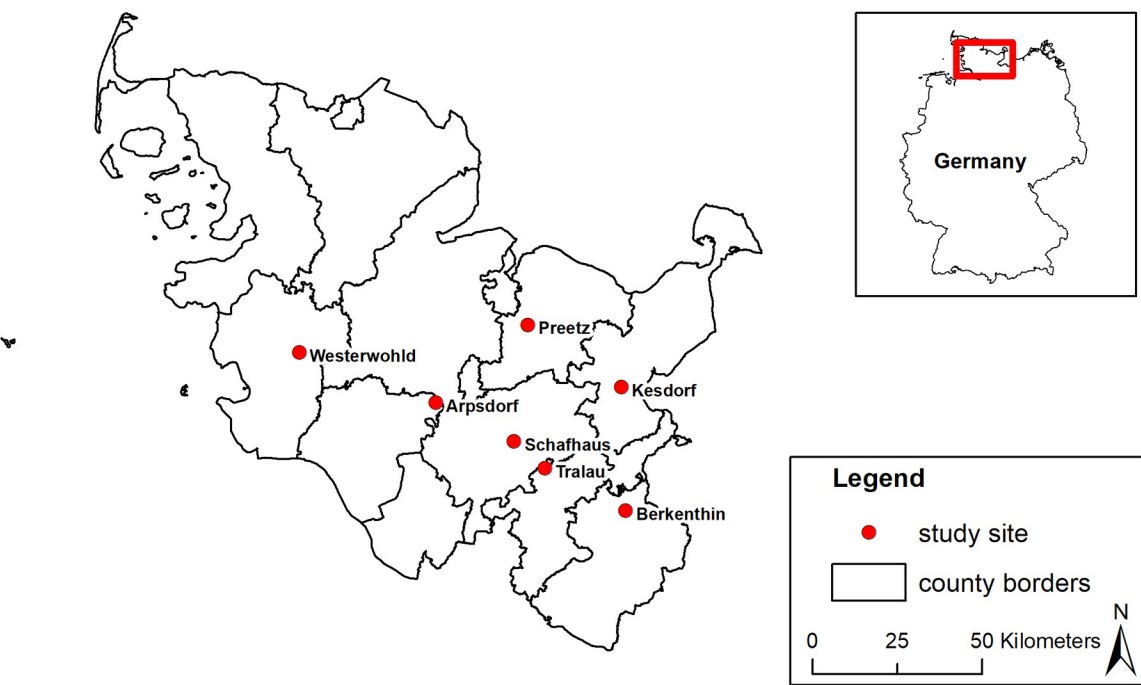

**Fig 1. Study area in Schleswig-Holstein, Northern Germany.**

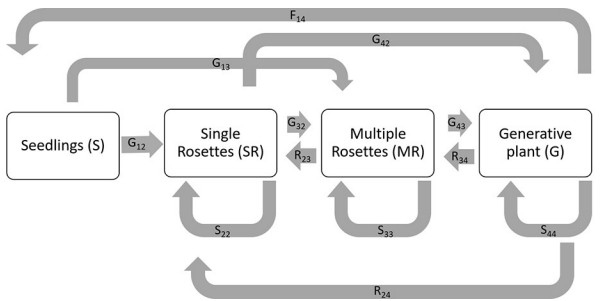

**Fig 2. Life cycle graph for *J. vulgaris* and the corresponding transition matrix.** Boxes indicate life stages, arrows represent the possible transitions between them, and letters show the connection between each transition and its matrix entry (F, fecundity; G, growth; S, stasis; R, retrogression).

individuals were assigned to one of three life stages: (i) seedling (S), (ii) rosettes (with one single rosette (SR) or multiple rosettes (MR)), and (iii) generative plants (G) with flower stalks. Altogether, 40 466 individuals for one or more transitions were monitored, summing up to 100 199 observations.

The initial species composition of the vegetation was recorded from June to August 2015. In subsequent years, vegetation assessment according to the Braun-Blanquet scale [36] took place between May and August. The Red List status of recorded species was noted [37]. Target species were defined as character species for species-rich grasslands in Northern Germany according to Schmidt et al. [38].

## Transition matrix model and population dynamics

Based on the frequency distribution of the recorded life stages, a 4x4-transition matrix (based on S, SR, MR, and G) was constructed for each study site, treatment, and year (Fig 2). Each matrix element ($a_{ij}$) was calculated from the number of individuals in stage $j$ in year $t$ that passed to stage $i$ in year $t + 1$, divided by the column total of stage $j$ [39]. Fecundity was determined by dividing the number of generative plants (G) in year $t$ by the number of seedlings (S) in the following year $t + 1$ [40].

The population growth rate was calculated by pooling all plants within each life stage for all study sites for each treatment and year. A 95% confidence interval was established by bootstrapping the data (5 000 iterations) for each treatment and year [39] in R [41]. For fecundity values, we used mean fecundities per treatment and year.

Life-table response experiments (LTREs) were conducted using matrices based on vital rates to analyze the contribution of different vital rates to the difference in the population growth rate ($\Delta\lambda$) between each treatment and the control [39]. Each matrix element is a product of the lower-level vital rates: survival ($\sigma_j$), stasis ($\gamma_{i = j}$), growth ($\gamma_{i > j}$), retrogression ($\gamma_{i < j}$), and reproduction ($\Phi_{ij}$) [42]. To compensate for differences in absolute values of $a_{ij}$, we analyzed elasticity, defined as the proportional change in $\lambda$ caused by a proportional change in $a_{ij}$ [42]. Elasticities sum to one and reflect the relative importance of matrix elements for population growth rate. All analyses were performed with the program POPTOOLS version 3.2 [43].

## Statistical analysis

To answer our first question regarding population growth (lambda $\lambda$) we used a mixed model approach [44, 45]. Fixed factors were treatment and year and study site was set as a random factor. The interaction effect of treatment and year was preliminarily tested to not be significant (or "to be statistically insignificant") based on a model with a pseudo factor representing a

mixture of treatment and year. Residuals were visually checked for normality and heterosce-dasticity. Based on this model, a Pseudo $R^2$ was calculated [46] and an analysis of variances (ANOVA) conducted, followed by multiple contrast tests [e.g., see 47] in order to compare the several levels of *treatment* and *year*, respectively.

To analyze the development of *numbers of plants and their life stages* we divided the dataset in separate blocks according to treatments and employed a one-way ANOVA with the factor year for every single treatment. For each study site, average numbers of plants per square-meter from the three plots per block were used. To identify differences between the treatments, years, and life stages, Tukey's HSD (honestly significant difference) post hoc test was applied.

For our second question about species diversity, we calculated species richness and evenness [48]. Species evenness was calculated by dividing the Shannon-index by the natural logarithm of species richness [48]. Species richness and evenness were analyzed within every single treatment with a one-factorial ANOVA with the factor year. To investigate whether species composition fluctuated more strongly in the treatments compared to the control plots, the temporal species turnover rate for each population and treatment was calculated as $(NR + D)/(n_t + n_{t+1})$ [49]. NR denotes the number of species per plot that were newly recorded in year $t + 1$ but did not occur on the plot in year $t$. D is the number of species that disappeared during the transition from year $t$ to year $t + 1$. $N_t$ and $n_{t+1}$ denote the species numbers in year $t$ and $t + 1$, respectively.

All statistical analyses were calculated using the statistical software R [41].

## Results

### Population dynamics of *J. vulgaris*

**Abundance and life stages.** The abundance of *J. vulgaris* individuals did not differ significantly between treatments; however, a trend was identified (ANOVA $F_{6, 204} = 1.955$ $P = 0.073$). Year had a significant effect on *J. vulgaris* abundance ($F_{4, 204} = 6.6478$, $P < 0.0001$).

The seven treatments led to three distinct response patterns (Fig 3). Both *biodiversity* treatments showed an initial decline in *J. vulgaris* abundance but a rebound after the fourth year. The *combination* treatments, the *seed cut* treatment, and the control showed an oscillating pattern with falling and rising *J. vulgaris* abundance every other year. Under all treatments, *J. vulgaris* abundance was on average lower in the fourth year of the study compared to the first year. Similarly, the abundance of rosettes and flowering plants was slightly lower at the end of the study in 2019 compared to its start in 2015 under all treatments ($52 \pm 8$ in 2019 vs. $60 \pm 5$ in 2015). Only the *flower cut* treatment led to a continuous decrease in *J. vulgaris* abundance, while statistically this was only a trend ($F_{4,30} = 2.558$, $P = 0.059$).

Fluctuations in *J. vulgaris* abundance in non-tilled sites between years were mainly driven by differences in the numbers of vegetative plants, while the number of flowering plants showed an almost continuous decrease during the study period (see Fig 4; ANOVA $F_{4,65} = 10.46$, $P < 0.0001$). This decrease was significant within *biodiversity* treatments, *seed cut*, and *combination 1* treatments.

**Population growth and life-table-response-experiments.** Population growth rates did not differ significantly between treatments ($F_{6,153} = 1.541$, $P = 0.168$). For the *flower cut* treatment, the population growth rate was significantly below one throughout the study period, indicating a constant decline in population size (Fig 5). All other treatments did not lead to a constant decline in population sizes but showed considerable variation between years. Interestingly, years with relatively high population growth rates were followed by years with lower population growth rates, which led to a significant year effect on λ (ANOVA $F_{3, 158} = 15.35064$, $P < 0.0001$). The substantial population growth setback in 2018 (year 17–18) was prominent.

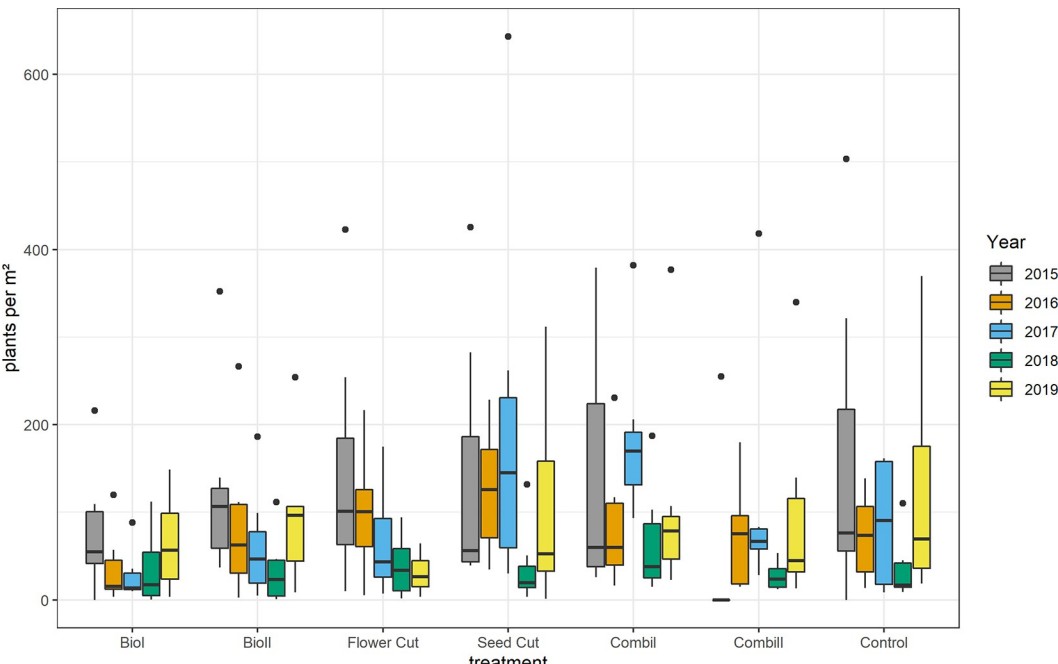

**Fig 3. Abundance of *J. vulgaris* plants per one square meter according to treatment and year.** In the box plots, the middle lines represent the median, boxes represent the first and third quartiles, lower and upper bars represent the minimum and the maximum, and points represent outliers (i.e. points above 1.5 SD). There were no significant differences between treatments (TukeyHSD, P ≤ 0.05).

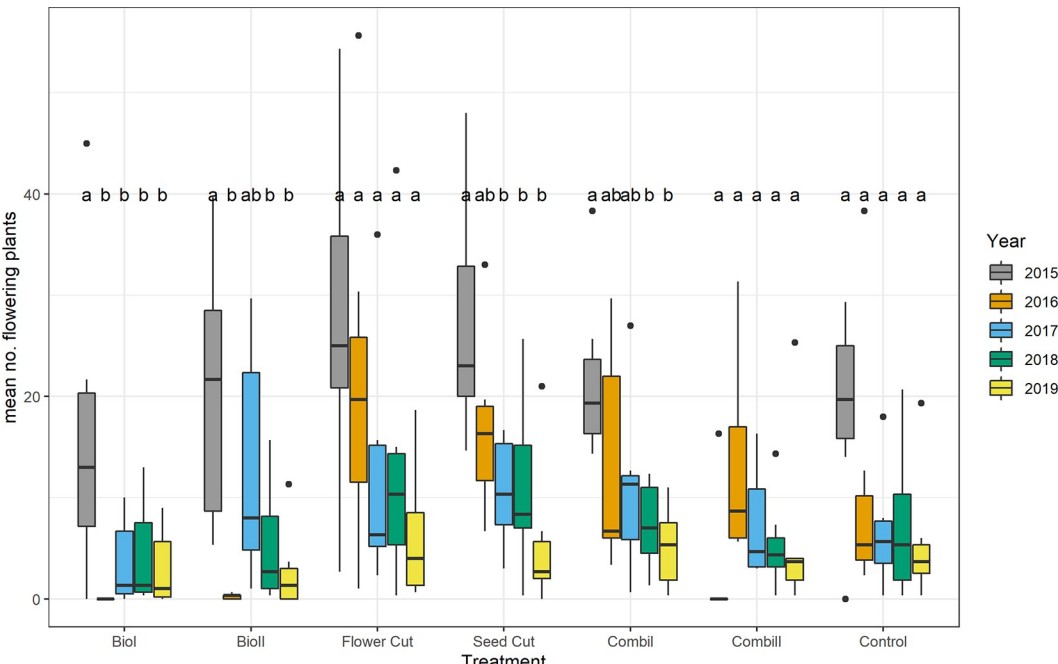

**Fig 4. Abundance of flowering *J. vulgaris* plants per one square meter according to treatment and year.** In the box plots, the middle lines represent the median, boxes represent the first and third quartiles, lower and upper bars represent the minimum and the maximum, and points represent outliers (i.e. points above 1.5 SD). Different letters indicate significant differences between the years within the treatments (TukeyHSD, P ≤ 0.05).

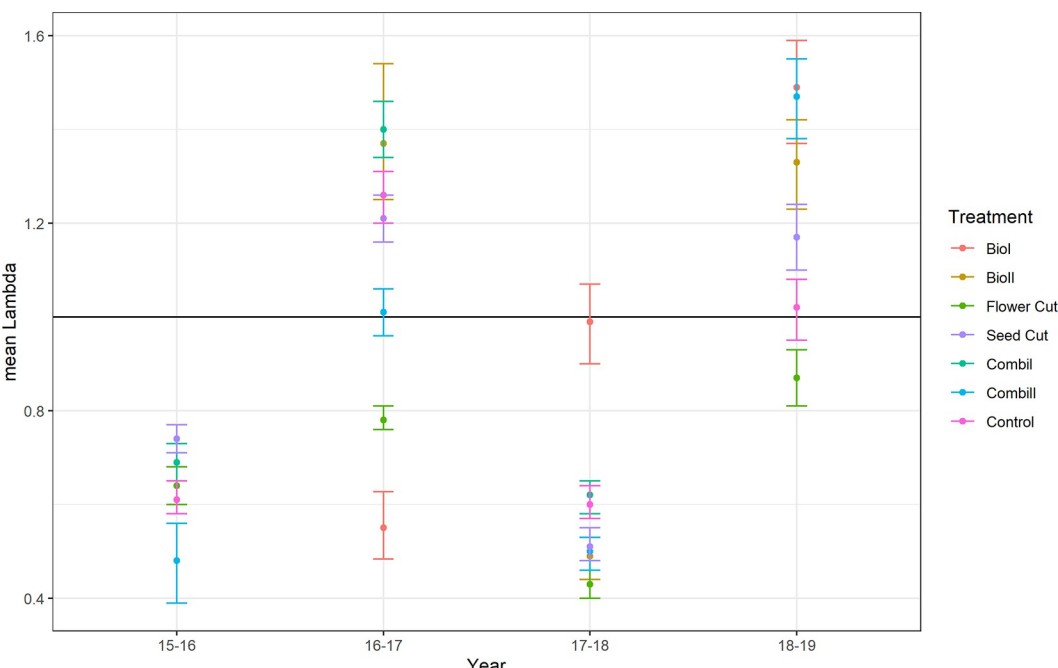

**Fig 5. Population growth rates for treatments according to years.** Shown are the means ± 95% CI. Treatments did not differ significantly. Constant population is marked with the line λ = 1.

An ANOVA of our mixed model including *treatment* and *year* as main factors and *study site* as a random factor revealed that *year* rather than *treatment*, mainly influenced the survival and population growth rates of *J. vulgaris*. In the dry year of 2018, density and abundance decreased but increased even more in the subsequent moist year of 2019. Consequently, there was a positive linear relationship between the amount of rainfall [50] and the abundance of seedlings found in the study plots (regression $y = -55.5360 + 1.0612*N$, $F_1 = 17.57$, $P < 0.0001$).

The LTRE analysis showed that the differences in the population growth rate (Δλ) between each treatment and the control were mainly the result of differences in generative reproduction and survival (Fig 6). The analysis also demonstrated that the influence of vital rates on population growth in the *flower cut* treatment was very different from all other treatments. Here stasis and retrogression have a considerable impact on positive population growth, as these vital rates are much higher than in the control.

The elasticity analysis revealed that three main transitions are crucial for population growth: i) the transition from seedling to single rosette ($G_{21}$), ii) from single rosette to flowering plant ($G_{24}$), and iii) from flowering plant to seedling ($F_{14}$). The importance of these transitions did not differ between treatment or year. All transitions contributed equally to population growth.

### Changes in vegetation composition

The overall mean number of vascular plants per one square meter was 23 ± 1 in 2015 and 29 ± 1 in 2019. Species richness differed between treatments (ANOVA $F_{6,210} = 3.204$, $P < 0.005$). The seed addition in *biodiversity* and *combination* treatments led to species enrichment. About five more plant species were found on plots with seed addition (*biodiversity* and *combination*) compared to control plots. In the last year of the experiment only species richness in the *combination* treatments was significantly higher than in the control treatment. All management measures except the control enhanced species richness (Fig 7).

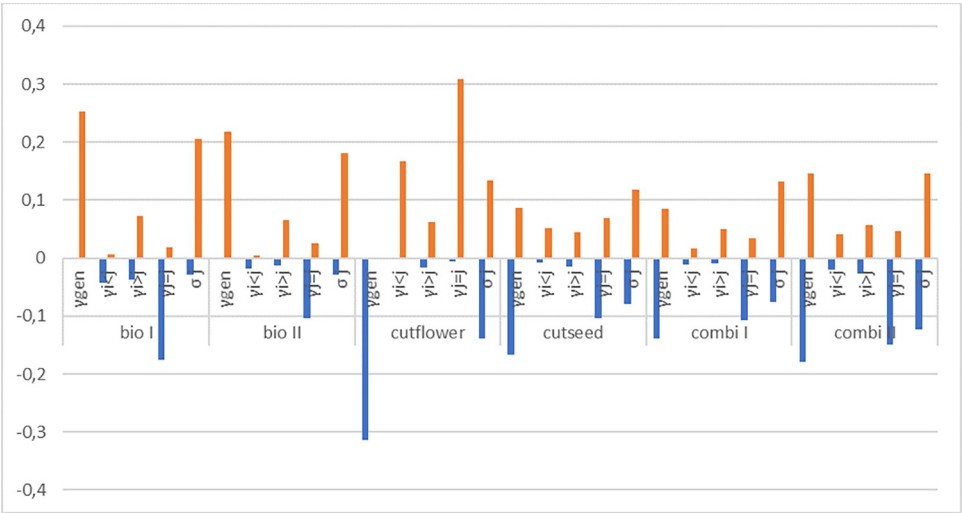

**Fig 6. Contribution of vital rates to the difference in population growth rate (Δλ) between the control and other treatments (bio I, *biodiversity 1*; bio II, *biodiversity 2*; cutflower, *flower cut* treatment; combi I, *combination* treatment 1; combi II, *combination 2*; cutseed, *seed cut* treatment) in the pooled *J. vulgaris* population, as determined by LTRE (life-table response experiment) analysis.** Bar sections above zero display the summed positive contributions (orange) and bar sections below zero the summed negative contributions (blue).

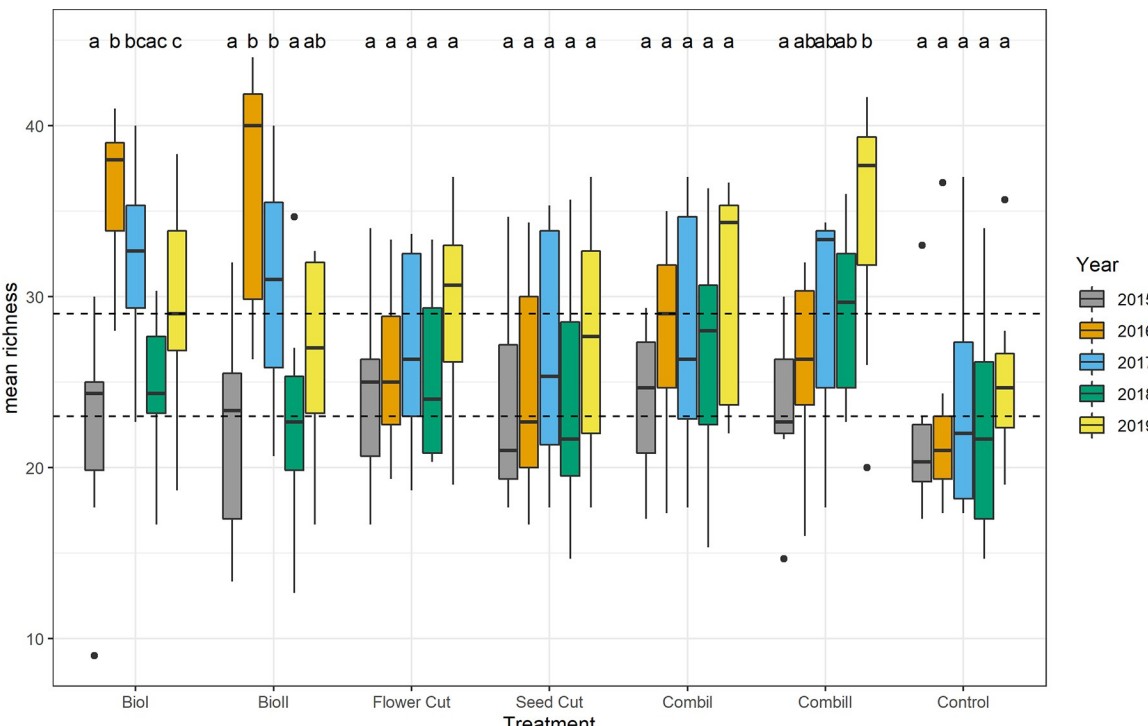

**Fig 7. Species richness according to year and treatment.** For explanation of the boxplot, refer to Fig 3. Different letters indicate significant differences (P ≤ 0.05; Tukey HSD) between years within one treatment. Dashed lines indicate mean species richness in 2015 (lower line) and 2019 (upper line).

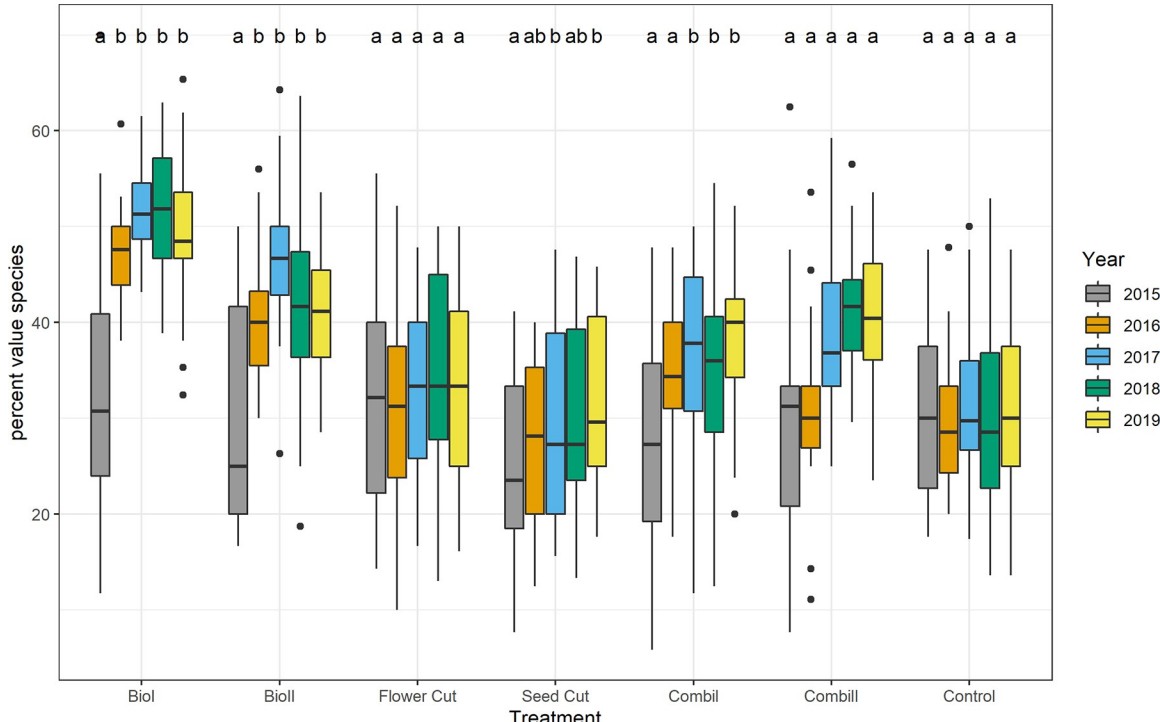

**Fig 8. Percentage of value species per year and treatment.** For explanation of the boxplot, refer to Fig 3. Different letters indicate significant differences (P ≤ 0.05; Tukey HSD) between years within one treatment.

Richness patterns varied over the study period (Fig 7) and were significantly different between treatments and years (ANOVA treatment $F_{6, 168} = 3.893$, $P < 0.005$; year $F_{3, 168} = 3.813$, $P < 0.05$). While in the first and second years after seed addition, the *biodiversity* treatments resulted in the highest species richness, in 2018 and 2019 the *combination* treatments were the most species rich. During all study years, the control treatment resulted in the lowest species richness, also only significantly different from the *biodiversity* treatments. Both cutting regimes had slightly but not significantly higher species richness compared to the control. Throughout the study period, species evenness was highest under treatments that included mowing (*flower* and *seed cut*, *combination 2*).

The number of red-listed plant species [37] differed significantly between treatments (ANOVA $F_{6, 189} = 31.08$, $P < 0.0001$). Most endangered plants occurred in the *biodiversity 1* treatment (mean: three endangered species per one square meter vs. zero or one species in the control). Plants categorized as characteristic grassland species were most numerous in the *biodiversity 1* treatment, followed by hay transfer (*biodiversity 2*) and *combination* treatments with mowing and sowing (Fig 8). *Biodiversity* treatments and the *combination 2* treatment led to significantly higher numbers of endangered plants than in the control.

Turnover rates of the vegetation differed significantly between treatments (ANOVA $F_{6, 140} = 16.74$, $P < 0.0001$) and were significantly highest under treatments with tilling in combination with seeding or plant material transfer, i.e. *biodiversity 1* and *2*, followed by *combination 2*, which combined to sowing pulses and mowing (S3 Fig). Turnover rates of pure cutting treatments, *combination 1*, and the control were significantly lower and did not differ from each other. The percentage of target species contributing to the turnover rate was highest for *biodiversity 1* (37 ± 0.02%) and *combination 2* (35 ± 0.03%) and lower for *combination 1* (29 ± 0.02%) even though these differences were not significant.

## Discussion

### Effects of regulation measures on *J. vulgaris*

Though we detected trends, the treatments did not significantly differ. *J. vulgaris* is a successional plant adapted to disturbance and quickly reacts to windows of opportunity being continuously created by grazing animals or because of drought. Moreover, as our treatments were designed to minimize unwanted harmful impacts on grassland ecosystems, we refrained from drastic measures. Nonetheless, our results reveal some promising approaches for *J. vulgaris* regulation that do not jeopardize nature conservation efforts.

The *biodiversity* treatments led to a decrease in *J. vulgaris* abundance in the first two to three years (Fig 3). The decrease was especially pronounced, significant, and lasting in the *biodiversity 1* treatment for the flowering stage (Fig 4). Although the general abundance decline in the *biodiversity* treatments after the first years was not significant, the results concur with previous findings that *J. vulgaris* densities decline as a result of sowing grassland species into the sward after soil disturbance [50, 51]. In our experiment, the control plots with plants that shed seeds were probably in too close together, which might have diluted treatment effects. As our study plant is wind dispersed and most seeds germinate within one year after seed shedding, it is very likely that upcoming seedlings originated from nearby sites rather than from the soil.

Declines in *J. vulgaris* abundance following tilling seem counterintuitive since establishment of *J. vulgaris* is favored by soil disturbance [9]. Consequently, soil disturbance was combined with sowing grassland species. Lawson et al. [2004] revealed a reduction in *J. vulgaris* densities in the first year after ploughing and sowing grassland species. Similarly, Bezemer et al. [51] reported reduced *J. vulgaris* abundance on plots that were sown with grassland species compared to control plots without seed addition over a period of eight years.

According to the biotic resistance theory [20], invasion vulnerability in species-rich systems should be lower than in species poor ones. While this might be true in the long run, at shorter temporal scales, species-rich plant communities may also be invaded if open soil patches remain that favor the establishment of light-demanding pioneers such as *J. vulgaris* [14]. Thus, we cannot confirm the results of Bezemer et al. [29] and Lawson et al. [52] that *J. vulgaris* density declines after ploughing and sowing, as we did not find lower *J. vulgaris* numbers in sown plots compared to untreated plots in the four years after tilling and sowing took place. We assume that this is due to differences in the initial setting of the study by Bezemer et al. [29] and our study. While in our study, grazed grassland was the control baseline, i.e. cattle caused continuous small-scale disturbances throughout the study, Bezemer et al. [29] began with an un-vegetated control, which was mown once a year. Repopulation of the early successional *J. vulgaris* might not have been prevented as around 33% higher bare ground cover compared to control plots (S4 Fig) offered enough establishment opportunities for the weed.

Furthermore, it is very probable that site conditions diluted management impacts. This phenomenon, i.e. spatial variation in the factors limiting plant populations can modulate treatment effects on plant population dynamics, was also shown by Rand et al. [53]. Thus, strategies for population management may have to pay attention to specific site conditions.

The LTRE analysis showed that the population growth on tilled sites was due to increased generative reproduction (Fig 5). The increase in *J. vulgaris* might also have occurred due to changes in the soil biota after tilling. While mature soil biota are thought to suppress *J. vulgaris* growth over time [50, 52, 54], soil disturbance might set soil biota back to an earlier developmental state, where negative effects on *J. vulgaris* are less pronounced. van de Voorde et al. [55] reported an increased growth of invaders in tilled soil due to changes in the soil community, which supports this assumption.

Besides tilling, mowing is a commonly used tool to control noxious weeds [19]. It can prevent seed production, reduce carbohydrate reserves, and give advantages to desirable perennial grasses. Our results show that the success of mowing depends on timing and frequency. While mowing once before seed dispersal failed to reduce the population growth rate or abundance of *J. vulgaris*, mowing twice shortly before the bloom resulted in a growth rate significantly and constantly below one and constantly decreased *J. vulgaris* abundance. This concurs with the finding that the optimal time for mowing noxious weeds is during the flowering stage before seed development [19]. This too is true for *J. vulgaris*, as the highest proportion of nutrients and energy is channeled into aboveground organs during bloom [58].

According to the pattern of population growth in the *flower cut* treatment, the LTRE analysis showed that the lower population growth rate in *flower cut* compared to the *control* treatment was mainly caused by the suppression of generative reproduction, the most important means of mass colonization for pioneer plants [1].

Additionally, repeated mowing leads to a denser sward and thus fewer open soil patches, which prevents the emergence of *J. vulgaris* seedlings. Furthermore, we found that continuous mowing twice a year for more than two years is needed to drive down *J. vulgaris* abundance. Otherwise, high numbers of plants either remain in their life stage or become rosettes so that prolific blooming will only be postponed.

In accordance with our results, Siegrist-Maag et al. [13] found that cutting *J. vulgaris* early and at least two times significantly weakens the plant. The LTRE analysis showed that impeding the fertility transition, i.e. the production of seeds, was the decisive property of the *flower cut* treatment causing *J. vulgaris* to decline, particularly in 2017, two years after the treatment began. The need to mow repeatedly over several years for successful weed control is also mirrored in the estimated survival rates in the *flower cut* treatment.

The LTRE-analysis showed that survival of *J. vulgaris* plants in the *flower cut* treatment in 2016/17 was higher than in the control. This is in accordance with the findings of Crawley et al. [56], who found that plants defoliated by the butterfly *Tyria jacobaeae* died less than those setting seeds, probably due to the higher resource investments involved in seed production compared to regrowth. Cumulative effects of constant mowing regimes were also found by other studies [57]. Increased retrogression and stasis of plants that are ready to bloom in the next year reflected this mechanism in our LTRE analysis.

Finding the optimal cutting date, after which *J. vulgaris* will not bloom again and produce vital seeds is difficult. Contrary to the effects of repeated cutting (*flower cut*), cutting once before seeds dispersed did not influence either population dynamics or *J. vulgaris* abundance. Even though more flowering plants completed their lifecycle and died, new seedlings from post-ripened seeds compensated for this mortality. In *J. vulgaris*, the first germinable seeds occur with the first withered capitula [33]. Eisele [33] suggested cutting *J. vulgaris* when 10% of the capitula begin to flower within half of the generative plants. However, under field conditions where the phenology of *J. vulgaris* is not synchronous, this stage is rather difficult to assess and is close to the emergence of the first germinable seeds. If cut too early, however, enough resources remain for vegetative reproduction [58].

Rosettes and seedlings constituted the largest proportion of *J. vulgaris* plants, which is in accordance with Siegrist-Maag et al. [13], who also found that 50% of *J. vulgaris* populations are made up of rosettes. Yet, *J. vulgaris* population structure and growth rates were highly variable across years, such that year had a stronger effect on population growth than management. We observed an oscillating pattern with falling and rising *J. vulgaris* abundance every other year in *seed cut*, *combination*, and *control* treatments (Fig 3). Extreme weather conditions, such as exceptionally hot or dry summers, affect population growth through immediate effects on vital rates [59, 60]. The low rainfall in 2018 caused a decline in the population growth rate

from 2017 to 2018 and a density decrease at all sites under all treatments. In accordance with our general observation of increased seedling numbers when rainfall was higher, we found that the population growth rate from 2018 to 2019 increased again after increased precipitation in 2019 (Fig 5). *J. vulgaris* populations probably profited from more light on the ground from 2018 to 2019, because the sward was less dense after the drought of 2018. Crawley et al. [56] found that plant recruitment was microsite limited and depended very much on weather and the activity of rabbits, boars, and moles, i.e. enough light and water for germination [14]. In our experiment, trampling and grazing from cattle can create windows of opportunity for *J. vulgaris* establishment [61].

Our analysis showed that all transitions, from seedling to rosette, from rosette to flowering plant, and from flowering plant to seedling contributed equally to population growth rate. This is in accordance with Dauer et al. [62], who found that transitions from rosette to flowering plant and from flowering plant to rosette were especially important for population growth.

## Development of species richness and vegetation composition

Botanical richness in our study either remained constant or increased. Thus, none of the measures applied to regulate *J. vulgaris* counteracted the goal of preserving species richness. The highest species richness occurred in the combination treatment of seeding and cutting (*combination 2*), where slightly different seed mixtures were sown in consecutive years and resident species and newly sown species coexisted, whereas species diversity in *control* treatments stayed lowest. *Combination 2* also reached high evenness-indices, as did the *flower cut* treatment. In general, evenness was significantly higher under mowing regimes (*flower cut*, *seed cut*, and *combination 2*), in which more species were able to establish due to reduced competition by the dominant grasses [63]. Diversity patterns of the treatments changed over time.

*Biodiversity* treatments were the most species rich in the first two years after tilling and seeding or green hay transfer. However, species numbers and cover of target species in *biodiversity* treatments declined after the third year (Figs 7 and 8). Other studies also showed that the initially high species numbers in sown treatments decreased after some years [51, 64, 65]. One reason for this decrease is that ruderal species and former arable weeds occur initially after tilling and vanish later on [27, 66]. Therefore, species turnover was highest in till-sow treatments (*biodiversity* treatments *1* and *2*) and seed-addition treatments (*combination* treatments *1* and *2*), whereas turnover in the pure cutting treatments (*flower* and *seed cut*) did not differ from the control.

Species turnover in the *biodiversity* and *combination* treatments led to a higher share of characteristic grassland species. The percentage of characteristic grassland species was highest in the *biodiversity* and *combination* treatments (Fig 8). As numerous other studies have shown, unless dispersal limitations are overcome by actively introducing seeds, it is rather unlikely that species-rich grasslands will develop on former arable land [67, 68]. This is due to the disappearance of characteristic grassland species in intensively used agricultural landscapes [69]. This is especially problematic within modern agricultural landscapes in Northern Germany, which are characterized by severe habitat fragmentation and biodiversity losses in grasslands [70].

The percentage of characteristic grassland species in *biodiversity* treatments was approximately 40%, which was 10% higher than in treatments without seed addition. While the number of species in the *biodiversity* treatments decreased again, the number of species in the *combination 2* treatment increased constantly over the study period and exceeded the number of species in the *biodiversity* treatments. This increase in species richness may have been supported by sowing different grassland species twice, which lowers the risk that establishment

fails due to unfavorable germination conditions when plants are sown in one year only. Besides the positive effect of two sowing applications, the establishment of higher species numbers in the *combination 2* treatment may have been supported by the yearly cut, which is known to favor the occurrence of forbs over grasses [71]. In the *control*, dominance of grasses and occurrence of ruderal species was higher than in all other treatments. This suggests that residual high soil fertility on ex-arable land may reduce the survival of characteristic grassland species [72], when the concurrence regime is not shifted via mowing.

The most promising method to promote threatened plants is tilling and sowing (*biodiversity 1*), probably because germination conditions for seeds in tilled plots are better than in non-tilled plots. In the latter, the accumulation of living and dead biomass and the encroachment of competitor species can hinder germination and establishment [66, 73]. However, there was also a noticeable effect of seed addition by hand–probably because the trampling of cattle can result in relatively large bare patches, potentially increasing the establishment opportunities for sown seeds [74]. Various studies [75, 76] have shown that gap size in the sward is an important element of successive seed introduction.

## Conclusion

Slight *J. vulgaris* decline occurred under all treatments and consequently we found no significant differences in *J. vulgaris* abundance between the applied treatments and the control. Even the most effective treatment, the *flower cut* treatment, which was significant by trend and the only treatment resulting in a population growth rate below one during the complete study period, did not lead to a complete disappearance of *J. vulgaris* but allowed on average five *J. vulgaris* plants per square meter to occur. Therefore, *J. vulgaris* management in nature conservation grasslands needs patience and will not eradicate the weed.

The *flower cut* treatment, which led to a constant decline in *J. vulgaris* resembles a traditional non-intensive meadow management technique, which supports high plant species and animal species richness. However, the two cuts that were employed for this treatment may harm invertebrates. These drawbacks must be weighed against the treatment's regulating effect on *J. vulgaris*. This might restrict the application of this measure to sites with high conflict potential.

## Supporting information

**S1 Data.**
(XLSX)

**S1 Table. Characteristics of the study sites.** Field properties (name, location, region, applied mower).
(DOCX)

**S2 Table. Applied seed mixture on biodiversity and combination treatments.** Herbs and grasses applied in moraine and hill land.
(DOCX)

**S1 Fig. Development of seedling numbers according to treatment and year.** In the box plots, middle lines represent the median, boxes represent the first and third quartiles, lower and upper bars represent the minimum and the maximum and points represent outliers (i.e. points above 1.5 SD). No significant differences were found.
(DOCX)

**S2 Fig. Development of rosette numbers according to treatment and year.** In the box plots, middle lines represent the median, boxes represent the first and third quartiles, lower and upper bars represent the minimum and the maximum and points represent outliers (i.e. points above 1.5 SD). No significant differences were found.
(DOCX)

**S3 Fig. Turnover rates for treatments.** In the box plots, middle lines represent the median, boxes represent the first and third quartiles, lower and upper bars represent the minimum and the maximum and points represent outliers (i.e. points above 1.5 SD). Different letters indicate significant differences between the treatments (TukeyHSD, P ≤ 0.05).
(DOCX)

**S4 Fig. Percentage of bare ground in plots.** In the box plots, middle lines represent the median, boxes represent the first and third quartiles, lower and upper bars represent the minimum and the maximum and points represent outliers (i.e. points above 1.5 SD). No significant differences were found.
(DOCX)

## Acknowledgments

We gratefully acknowledge Diethart Matthies for his help with the bootstrap analyses and Mario Hasler for statistical advice. Furthermore, we thank Hollyn Hartlep and David S. Bennett for proofreading. We also want to thank Henning Nissen for his expertise in creating maps.

## Author Contributions

**Conceptualization:** Tim Diekötter, Tobias W. Donath.

**Data curation:** Henrike Wiggering, Tobias W. Donath.

**Formal analysis:** Henrike Wiggering, Tobias W. Donath.

**Funding acquisition:** Tim Diekötter, Tobias W. Donath.

**Investigation:** Henrike Wiggering.

**Methodology:** Henrike Wiggering, Tobias W. Donath.

**Project administration:** Henrike Wiggering, Tobias W. Donath.

**Resources:** Henrike Wiggering.

**Supervision:** Tim Diekötter, Tobias W. Donath.

**Validation:** Tobias W. Donath.

**Visualization:** Henrike Wiggering.

**Writing – original draft:** Henrike Wiggering.

**Writing – review & editing:** Henrike Wiggering, Tim Diekötter, Tobias W. Donath.

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
