## [Decision Letter · Decision Letter 0]

1 Apr 2021

PONE-D-21-05618

Regulation of Jacobaea vulgaris by varied cutting and restoration measures

PLOS ONE

Dear Dr. Möhler,

Thank you for submitting your manuscript to PLOS ONE. After careful consideration, we feel that it has merit but does not fully meet PLOS ONE’s publication criteria as it currently stands.** In general, ****the manuscript needs an in-depth review with special attention to methodology**.  Therefore, we invite you to submit a revised version of the manuscript that addresses the points raised during the review process.

We look forward to receiving your revised manuscript.

Kind regards,

Jose L. Gonzalez-Andujar

Academic Editor

PLOS ONE

Additional Editor Comments:

Describe in M&M the elastictiy analysis performed

Journal Requirements:

2. We note that Figure 1 in your submission contain [map/satellite] images which may be copyrighted. All PLOS content is published under the Creative Commons Attribution License (CC BY 4.0), which means that the manuscript, images, and Supporting Information files will be freely available online, and any third party is permitted to access, download, copy, distribute, and use these materials in any way, even commercially, with proper attribution. For these reasons, we cannot publish previously copyrighted maps or satellite images created using proprietary data, such as Google software (Google Maps, Street View, and Earth). For more information, see our copyright guidelines: http://journals.plos.org/plosone/s/licenses-and-copyright.

You may seek permission from the original copyright holder of Figure 1 to publish the content specifically under the CC BY 4.0 license. 

If you are unable to obtain permission from the original copyright holder to publish these figures under the CC BY 4.0 license or if the copyright holder’s requirements are incompatible with the CC BY 4.0 license, please either i) remove the figure or ii) supply a replacement figure that complies with the CC BY 4.0 license. Please check copyright information on all replacement figures and update the figure caption with source information. If applicable, please specify in the figure caption text when a figure is similar but not identical to the original image and is therefore for illustrative purposes only.

Reviewers' comments:

Reviewer's Responses to Questions

**Comments to the Author**

1. Is the manuscript technically sound, and do the data support the conclusions?

Reviewer #1: Yes

Reviewer #2: Partly

2. Has the statistical analysis been performed appropriately and rigorously? 

Reviewer #1: I Don't Know

Reviewer #2: N/A

3. Have the authors made all data underlying the findings in their manuscript fully available?

Reviewer #1: Yes

Reviewer #2: Yes

4. Is the manuscript presented in an intelligible fashion and written in standard English?

Reviewer #1: Yes

Reviewer #2: Yes

5. Review Comments to the Author

Reviewer #1: The authors have compared six different management regimes for reducing the abundance of Jacobaea vulgaris. These regimes included combinations of seeding, soil disturbance, and mowing to facilitate natural communities while suppressing the weed. They found that seeding increased resident community metrics, and cutting before bloom (twice) reduced the weed’s population growth rate. However, no treatment significantly reduced abundance of J. vulgaris relative to controls, though flowering emerges as a stable/detectable management target. The experiment replicates a range of treatment possibilities across several sites, and follows them for multiple years. Thus, they were able to detect certain trends that may prove useful. I appreciate the attention to weed vital rates as well as abundance in the evaluation of different treatments, and the examination of community transitions as a response variable. Vital rate estimates data are important to quantifying weed invasions, but are often not considered. Overall, I liked the study and the approach provides comparisons to tease apart advantages of different methods that apply similar pressure on the weed, and think it will be informative and of interest to a broad audience. Portions of the methods require some clarification; it’s not completely clear to me at times what is being compared and whether treatments are compared within or across sites. However, I don't fear I would have issues with the general structure of the analysis. There’s also a few comments that need more context. With a thoughtful revision of the manuscript, I think it would be a solid contribution to the literature. Hopefully the questions and comments below will be useful in this.

Questions:

Would you define “milling”? I’m not familiar with this practice in the context of soil disturbance.

Though you detected trends, the treatments didn’t significantly affect J. vulgaris. I was wondering about site factors that may have interacted with management to dilute impacts. You control for “site” as a random factor in population growth (L195-196), but it’s not clear that an interaction between site and treatment has been considered.

-Do you think site differences in pressures affecting recruitment and transitions among life stages may lead to different advantages of treatments among sites? (for instance, see Rand et al. 2020 Oecologia 193:143-153) Some of the concepts you reference around precipitation in the discussion (e.g., L410-412) would also be applicable to this question.

-Because it’s wind dispersed, do you think proximity of blocks might muddy treatment effects on seedling recruitment? That is, were they sufficiently separated to minimize the invasion/seed dispersal from adjacent plots?

L41: Should this be overall plant diversity? As it includes some introduced seed, some ruderal weeds, etc.

L133- each treatment was implemented over a 90 m2 area, so each site consisted of seven 90 m2 blocks?

L148: How did you choose additional seed to mix?

L152: Seems out of place. Should this go up to 149-150?

L179: I’m not sure where the pooling is occurring. I assume you mean that each site had a population growth rate in each treatment and year, and you’re pooling your three plots within each block to each treatment?

189 (analysis): may be helpful to organize around the two questions in the intro. I’m not sure exactly what is being tested at times (among years, treatments, at what level replication occurs). For instance, is site treated as a random factor in comparing the number of plants among treatments? (L201-205)

Confusion regarding ANOVAS and what’s tested (e.g., L234-5)

L249 (population results): why not do mixed models with population growth? (site effects/interactions)

L254-257: high then low growth rates: do you think this emerges from seedling flushes or from more flowering adults? Just curious.

L260-263: Not sure where this rainfall data and linear analysis comes from? Please add to supplement, or provide a reference.

L421: Do you think that even if rosette-rosette is not be critical to growth, it might maintain a potential pool through time that allows populations to persist even when not growing (L272-274)? Certainly the above suggests there’s a holdover in flowering and expansion in droughty conditions.

L353: Does milling eliminate standing cover?

L430: Soil disturbance doesn’t seem as beneficial to site quality improvement, sowing and mowing (more frequently) does

L464: Why soil fertility? Can you add a couple of sentences here to provide context for this assertion?

Figures and Tables:

Table 1: may be helpful to categorize treatments to highlight comparisons, e.g., columns: disturbance|frequency of disturbance|seeding|frequency of seeding| seed mix. Especially as hand sowing seemed to make a difference, but not drilling (L470)

L174: should this table have a legend? And should retrogression be included? L177 leaves out reference to “R”, which is included in both the figure and table. (Figure 2: Is “R” retrogression?)

Figure 6: Bar colors for each category? (L277: bars are treatment and controls?)

Check that figure categories are ordered consistently - Figures 3 and 8 are ordered differently.

A few minor typos questions:

L38 “Its control…”

L114: “vegetative” bud?

L126-7: “…very high with an (?) average….at the experiment’s start”

L324: sowing in pulses? Or sowing pulses?

L338: no comma after milling

L385: orphaned “r” after “…causing…”

L399: …compensated for this mortality

Reviewer #2: The work is interesting but presents problems as can be seen in the attached text.

Negative points: First, the work found no differences for the treatments tested, but built a conclusion as if it had found big differences. Therefore, reformulate the writing of the text based on the data found and do not make conclusions that the data do not support.

Figures need better presentation, especially figures 5 and 6.

Very long conclusion and nothing objective (it is not the time for discussion). It needs more objectivity.

Restructure the text according to the data found.

6. PLOS authors have the option to publish the peer review history of their article (what does this mean?). If published, this will include your full peer review and any attached files.

Reviewer #1: No

Reviewer #2: No

---

## [Author Response · Author response to Decision Letter 0]

25 Nov 2021

Dear Editor and Reviewers,

We thank you very much for your valuable efforts and helpful comments and critiques. Based on both reviews we created an updated and improved manuscript.

Further down, we would like to directly address discussed points, with your critique stated repeated (following R1/R1) and our answer (following A). Minor points were track changed directly in the manuscript.

In any case, we thank you very much for taking our manuscript into consideration anew and thank you again for your helpful critique and comments.

Yours sincerely

Henrike Wiggering (birthname Möhler)

---

## [Decision Letter · Decision Letter 1]

25 Jan 2022

PONE-D-21-05618R1Regulation of Jacobaea vulgaris by varied cutting and restoration measuresPLOS ONE

Dear Dr. Wiggering,

Thank you for submitting your manuscript to PLOS ONE. After careful consideration, we feel that it has merit but does not fully meet PLOS ONE’s publication criteria as it currently stands. Therefore, we invite you to submit a revised version of the manuscript that addresses the points raised during the review process.

We look forward to receiving your revised manuscript.

Kind regards,

Jose L. Gonzalez-Andujar

Academic Editor

PLOS ONE

Journal Requirements:

Additional Editor Comments (if provided):

The manuscript is acceptable for publication but it is necessary to make some minor modification for clarity and context as pointed out by reviewer #2.

Reviewers' comments:

Reviewer's Responses to Questions

**Comments to the Author**

1. If the authors have adequately addressed your comments raised in a previous round of review and you feel that this manuscript is now acceptable for publication, you may indicate that here to bypass the “Comments to the Author” section, enter your conflict of interest statement in the “Confidential to Editor” section, and submit your "Accept" recommendation.

Reviewer #1: (No Response)

Reviewer #2: All comments have been addressed

2. Is the manuscript technically sound, and do the data support the conclusions?

Reviewer #1: Yes

Reviewer #2: Yes

3. Has the statistical analysis been performed appropriately and rigorously? 

Reviewer #1: Yes

Reviewer #2: Yes

4. Have the authors made all data underlying the findings in their manuscript fully available?

Reviewer #1: Yes

Reviewer #2: Yes

5. Is the manuscript presented in an intelligible fashion and written in standard English?

Reviewer #1: Yes

Reviewer #2: No

6. Review Comments to the Author

Reviewer #1: The manuscript is improved and has addressed many of the reviewer comments. However, the authors still have a few issues with consistency (e.g., fecundity versus fertility [L169-170 & L179-180, L395], L201 should be second question, not third) and adequately describing methods for reported results (e.g., "elasticity" appears in the results (L277) but it is the first time this term is used, L202: how was evenness quantified (Pielou's index? Simpson's?, etc.). Though these examples are common metrics in ecological studies, they are important to define and contextualize. Overall, I think with some minor revisions for clarity and context, it could be acceptable for publication.

Reviewer #2: Dears,

In my opinion, the last detail to be corrected in the work is English, send it for review.

Regards.

7. PLOS authors have the option to publish the peer review history of their article (what does this mean?). If published, this will include your full peer review and any attached files.

Reviewer #1: No

Reviewer #2: No

---

## [Author Response · Author response to Decision Letter 1]

8 Mar 2022

Dear Editor and Reviewers,

We thank you very much for your time, efforts and helpful comments and critiques. Taking all of it into account we created an updated and improved manuscript.

We checked references anew and found no inconsistencies.

As proposed by reviewer 1 we replaced fertility with fecundity [L169-170 & L179-180, L395], corrected the numbering of questions in L201 and described methods we used for the calculation of "elasticity” and “evenness”, which was indeed an essential we forgot to mention in the first place. Thanks for the advice!

Furthermore, we had a second native English speaker check the manuscript and hope that it is in an understandable adequate language now.

We thank you for taking our manuscript into consideration anew and thank you again for your helpful critique and comments.

Yours sincerely

Henrike Wiggering, Tobias Donath, Tim Diekötter

---

## [Decision Letter · Decision Letter 2]

23 Mar 2022

PONE-D-21-05618R2Regulation of Jacobaea vulgaris by varied cutting and restoration measuresPLOS ONE

Dear Dr. Wiggering,

Thank you for submitting your manuscript to PLOS ONE. After careful consideration, we feel that it has merit but does not fully meet PLOS ONE’s publication criteria as it currently stands. Therefore, we invite you to submit a revised version of the manuscript that addresses the points raised during the review process. Please make the few minor changes suggested by reviewer 1.

We look forward to receiving your revised manuscript.

Kind regards,

Remigio Paradelo Núñez

Academic Editor

PLOS ONE

Journal Requirements:

Additional Editor Comments (if provided):

Dear authors,

Please make the few minor changes suggested by reviewer 1.

Reviewers' comments:

Reviewer's Responses to Questions

**Comments to the Author**

1. If the authors have adequately addressed your comments raised in a previous round of review and you feel that this manuscript is now acceptable for publication, you may indicate that here to bypass the “Comments to the Author” section, enter your conflict of interest statement in the “Confidential to Editor” section, and submit your "Accept" recommendation.

Reviewer #1: (No Response)

2. Is the manuscript technically sound, and do the data support the conclusions?

Reviewer #1: Yes

3. Has the statistical analysis been performed appropriately and rigorously? 

Reviewer #1: Yes

4. Have the authors made all data underlying the findings in their manuscript fully available?

Reviewer #1: (No Response)

5. Is the manuscript presented in an intelligible fashion and written in standard English?

Reviewer #1: Yes

6. Review Comments to the Author

Reviewer #1: The authors have adequately addressed issues raised, and I recommend acceptance. I've noted some minor technical tweaks that need resolution below (and one just aside comment that is purely due to interest, not issue). Thank you for the opportunity to read your work.

L142: how many weeks? an approximate or minimum number would suffice if the exact is unknown. But this would be necessary to recreate the experiment.

L192: you use the symbol for 'lambda' throughout the text, so please include it in the parenthetical aside

L312: please cite the index for red listed species

L348-9: interesting: how long might seeds lie dormant? Perhaps a combination of seedbank and wind-dispersal

L418: Eisele reference needs a year.

7. PLOS authors have the option to publish the peer review history of their article (what does this mean?). If published, this will include your full peer review and any attached files.

Reviewer #1: No

---

## [Author Response · Author response to Decision Letter 2]

5 May 2022

We added the minimum number of weeks in which the biodiversity treatments were fenced.

We added the lamda symbol in line 192.

We cited the Red List index used in the methods section (line 161, Mierwald 2006) and added the citation in line 312.

It is an interesting question how long seeds lie dormant. From our own experience there is no dormancy as we found new seedlings very soon after seed shedding. Moreover, we had a Bachelor Thesis treating the topic of the relevance of the seed bank and found that there were relatively little seeds in the seed bank but most seedlings resulted from the seed rain. Thus, we would expect that seed rain is more relevant than regeneration via seeds from the seed bank. However, it is very likely that both seedbank and wind-dispersal are contributing to new seedlings. 

We completed the citation of Eisele in line 418.

We thank you for taking our manuscript into consideration anew and thank you again for your helpful critique and comments.

---

## [Editor Report · Decision Letter 3]

10 May 2022

Regulation of Jacobaea vulgaris by varied cutting and restoration measures

PONE-D-21-05618R3

Dear Dr. Wiggering,

We’re pleased to inform you that your manuscript has been judged scientifically suitable for publication and will be formally accepted for publication once it meets all outstanding technical requirements.

Kind regards,

Remigio Paradelo Núñez

Academic Editor

PLOS ONE
---

## [Editor Report · Acceptance letter]

13 May 2022

PONE-D-21-05618R3 

Regulation of *Jacobaea vulgaris* by varied cutting and restoration measures 

Dear Dr. Wiggering:

I'm pleased to inform you that your manuscript has been deemed suitable for publication in PLOS ONE. Congratulations! Your manuscript is now with our production department. 

Kind regards, 

on behalf of

Dr. Remigio Paradelo Núñez 

Academic Editor

PLOS ONE